# Post-Resuscitation Partial Pressure of Arterial Carbon Dioxide and Outcome in Patients with Out-of-Hospital Cardiac Arrest: A Multicenter Retrospective Cohort Study

**DOI:** 10.3390/jcm11061523

**Published:** 2022-03-10

**Authors:** Nobunaga Okada, Tasuku Matsuyama, Yohei Okada, Asami Okada, Kenji Kandori, Satoshi Nakajima, Tetsuhisa Kitamura, Bon Ohta

**Affiliations:** 1Department of Emergency Medicine, Kyoto Prefectural University of Medicine, Kyoto 602-8566, Japan; kame0413oka.jin@gmail.com (T.M.); satoshi.nakajima2.9@gmail.com (S.N.); ohta2010@wind.ocn.ne.jp (B.O.); 2Department of Preventive Services, School of Public Health, Kyoto University, Kyoto 606-8501, Japan; yokada-kyf@umin.ac.jp; 3Department of Primary Care and Emergency Medicine, Graduate School of Medicine, Kyoto University, Kyoto 606-8501, Japan; 4Department of Emergency Medicine and Critical Care, Japanese Red Cross Society Kyoto Daini Hospital, Kyoto 602-8026, Japan; asami.tsuji0515@gmail.com (A.O.); knj.kandori@gmail.com (K.K.); 5Division of Environmental Medicine and Population Sciences, Department of Social and Environmental Medicine, Graduate School of Medicine, Osaka University, Osaka 565-0871, Japan; lucky_unatan@yahoo.co.jp

**Keywords:** out-of-hospital cardiac arrest, post-cardiac arrest care, blood gas analysis, carbon dioxide, hypercapnia, critical care outcomes

## Abstract

We aimed to estimate the association between PaCO_2_ level in the patient after out-of-hospital cardiac arrest (OHCA) resuscitation with patient outcome based on a multicenter prospective cohort registry in Japan between June 2014 and December 2015. Based on the PaCO_2_ within 24 h after return of spontaneous circulation (ROSC), patients were divided into six groups as follows: severe hypocapnia (<25 mmHg), mild hypocapnia (25–35 mmHg,), normocapnia (35–45 mmHg), mild hypercapnia (45–55 mmHg), severe hypercapnia (>55 mmHg), or exposure to both hypocapnia and hypercapnia. Multivariate logistic regression analysis was conducted to calculate the adjusted odds ratios (aORs) and 95% confidence interval (CI) for the 1-month poor neurological outcome (Cerebral Performance Category ≥ 3). Among the 13,491 OHCA patients, 607 were included. Severe hypocapnia, mild hypocapnia, severe hypercapnia, and exposure to both hypocapnia and hypercapnia were associated with a higher rate of 1-month poor neurological outcome compared with mild hypercapnia (aORs 6.68 [95% CI 2.16–20.67], 2.56 [1.30–5.04], 2.62 [1.06–6.47], and 5.63 [2.21–14.34], respectively). There was no significant difference between the outcome of patients with normocapnia and mild hypercapnia. In conclusion, maintaining normocapnia and mild hypercapnia during the 24 h after ROSC was associated with better neurological outcomes than other PaCO_2_ abnormalities in this study.

## 1. Introduction

Out-of-hospital cardiac arrest (OHCA) is associated with high mortality and poor neurological outcomes [1,2]. Finding appropriate post-resuscitation care to reduce the degree of brain injury after the return of spontaneous circulation (ROSC) from cardiac arrest is important for resuscitation science. In patients after ROSC, partial pressure of arterial carbon dioxide (PaCO_2_) abnormalities such as high PaCO_2_ due to lack of ventilation during cardiopulmonary arrest and cardiac or respiratory complications, and low PaCO_2_ due to excessive mechanical ventilation for resuscitation are common [3,4]. In brain-injured patients, including OHCA patients, some mechanisms including PaCO_2_ may impact cerebral blood flow and perfusion, thus, maintaining PaCO_2_ could help deliver better neurological outcomes. 

Recent guidelines by the European Resuscitation Council [5] and the American Heart Association [6] said that maintaining normocapnia (PaCO_2_ 35–45 mmHg) may be a reasonable goal in post-cardiac arrest care. A recent meta-analysis study by McKenzie et al. [7] showed that both hypocapnia and hypercapnia were associated with worse survival outcomes; on the other hand, some studies [8,9,10] reported that mild hypercapnia had better neurological outcomes compared with normocapnia. The controlled PaCO_2_ cutoff values vary among studies, and although we know about the approximate prognostic PaCO_2_ values, there are no studies with detailed PaCO_2_ classifications. Therefore, the optimal PaCO_2_ target remains controversial. 

Although there are studies that have classified PaCO_2_ values at 24 h after ROSC into three categories (hypocapnia, normocapnia, and hypercapnia) and evaluated their association with neurological outcomes, there are few randomized controlled trials and observational studies that have finely classified PaCO_2_ values and evaluated in more detail. This study aimed to estimate the target range of PaCO_2_ exposure with the better outcome by dividing patients with ROSC from OHCA into six categories including mild hypercapnia according to the PaCO_2_ values they were exposed to.

## 2. Materials and Methods

This study was a post hoc analysis of the Japanese Association for Acute Medicine (JAAM) OHCA registry [1] between June 2014 and December 2015. The JAAM-OHCA registry is a nationwide, multicenter prospective registry that included 56 institutions in 2014 and 73 institutions in 2015. The registry data included both pre- and in-hospital data; pre-hospital data were obtained from the All Japan Utstein Registry of the Fire and Disaster Management Agency [11], and in-hospital data were collected via an Internet-based system by physicians or medical staff at each institution. The details of the study protocol were previously described [1]. All methods were performed in accordance with the relevant guidelines and regulations. This registry and the study protocol were approved by the ethics committee of Kyoto Prefectural University of Medicine (approval ID: ERB-C-650-1) and each institution. The ethics committee waived the need for individual written informed consent.

### 2.1. Participants

Our study inclusion criteria were as follows: patients with ROSC after OHCA aged 18 years and over, patients in the protocol of assessing arterial blood gas (ABG), and disorders of consciousness at hospital arrival (defined as Glasgow coma score (GCS) motor <6). Patients in whom the main cause of OHCA was external including trauma or hangings, and those without PaCO_2_ data during 24 h post-ROSC were excluded.

### 2.2. Data Collection

PaCO_2_ values were collected three times; immediately after ROSC, upon admission to the intensive care unit (ICU), and 24 h after ROSC. In patients with pre-hospital ROSC, the data on PaCO_2_ immediately after ROSC used the first PaCO_2_ values obtained after hospital arrival or treated them as missing. Patient characteristics and pre- and in-hospital data were defined as sex, age, initial cardiac rhythm (ventricular fibrillation/pulseless ventricular tachycardia (VF/pVT), pulseless electrical activity (PEA), asystole, and other including after ROSC), witnessed the arrest, presence of bystander performed cardiopulmonary resuscitation (CPR), CPR duration, CGS at hospital arrival, hyperoxia exposure (≥PaO_2_ 300 mmHg) [12], treatments performed after hospital arrival (including using mechanical circulatory device (extracorporeal membrane oxygen and/or intra-aortic balloon pumping) and targeted temperature management (TTM)), cause of cardiac arrest (respiratory diseases, cerebrovascular diseases, malignant tumor, others, or unknown), 1-month mortality, and 1-month neurological outcome using the Glasgow-Pittsburgh cerebral performance category scale (CPC; 1 = good cerebral performance, 2 = moderate cerebral disability and independent in activities of daily life, 3 = severe cerebral disability and dependent on others for daily support, 4 = vegetative state, 5 = death/brain death) [13].

### 2.3. Outcome

The primary outcome was a poor neurological outcome (CPC ≥ 3) at 1 month after ROSC. The secondary outcome was 1-month mortality.

### 2.4. Statistical Analysis

We determined whether patients were exposed to hypocapnia and hypercapnia during the first 24 h after ROSC using ABG data from two (upon admission to the ICU and 24 h after ROSC) of three timepoints. PaCO_2_ immediately after ROSC was considered difficult to control clinically due to the direct influence of cardiac arrest and CPR and was treated as a separate independent variable. We defined severe hypocapnia as PaCO_2_ < 25 mmHg, mild hypocapnia as 25 mmHg ≤ PaCO_2_ < 35 mmHg, normocapnia as 35 mmHg ≤ PaCO_2_ ≤ 45 mmHg, mild hypercapnia as 45 mmHg < PaCO_2_ ≤ 55 mmHg, and severe hypercapnia as PaCO_2_ > 55 mmHg by thresholds based on previous studies about PaCO_2_ [9,10,14,15,16,17]. Patients were allocated into six groups as follows: severe hypocapnia exposure (one or more severe hypocapnic episode), mild hypocapnia exposure (one or more mild hypocapnic episode), normocapnia exposure (only normocapnia recorded), mild hypercapnia exposure (one or more mild hypercapnic episode), severe hypercapnia exposure (one or more severe hypercapnic episode), and both hypocapnia and hypercapnia exposure. Patients exposed to both mild and severe CO_2_ abnormalities, as shown by the ABG analysis conducted at two different time points, were included in the more severe group (e.g., a patient showing mild hypocapnia upon admission to the ICU but severe hypocapnia 24 h after ROSC was included in the severe hypocapnia group).

Patient characteristics, pre- and in-hospital data, and outcomes for the categories were compared using the Mann–Whitney U-test for continuous variables and the Chi-squared test or Fisher’s exact test for categorical variables.

We calculated adjusted odds ratios for outcomes using a multivariate logistic regression model to estimate the association between each PaCO_2_ exposure group and patient outcomes, adjusted for sex, age (18–64 years, 65–74 years, ≥75 years), witnessed arrest, bystander performed CPR, initial cardiac rhythm, CPR duration >10 min, GCS at hospital arrival (3 to 14), hyperoxia exposure, using the mechanical circulatory device, TTM, cause of cardiac arrest (acute coronary syndrome, cardiac cause excluding acute coronary syndrome (presumed cardiac cause), respiratory cause, cerebrovascular cause, malignant tumor, others, or unknown), PaCO_2_ immediately after ROSC (severe hypocapnia, mild hypocapnia, normocapnia, mild hypercapnia, severe hypercapnia), and PaCO_2_ group (severe hypocapnia exposure, mild hypocapnia exposure, normocapnia exposure, mild hypercapnia exposure, severe hypercapnia exposure, both hypocapnia and hypercapnia). We treated missing data for each variable as an “unknown category.” Potential confounding variables were selected and classified according to the models defined in the previous studies [18,19,20] and clinically important variables were added. In addition, we conducted a sensitivity analysis for all OHCA patients included in the study, after excluding patients with treatment using a mechanical circulatory device because the inclusion of the patients with the treatment, which may have a poorer outcome, may affect the results of the analysis. PaCO_2_ level immediately after ROSC is important in assessing patient prognosis, since it is thought to affect PaCO_2_ level for the next 24 h (especially PaCO_2_ upon admission to the ICU). Therefore, we also showed the patients’ character by PaCO_2_ levels immediately after ROSC and examined the relationship between PaCO_2_ immediately after ROSC and PaCO_2_ upon admission to the ICU.

All tests were two-tailed and a *p*-value < 0.05 was considered statistically significant. All statistical analyses were performed using JMP 14.0 (SAS Institute, Cary, NC, USA).

## 3. Results

### 3.1. Population

A total of 13,491 OHCA patients were registered during the study period. After excluding 3216 patients not included in the ABG assessment protocol, 263 patients aged <18 years, 6496 non-ROSC patients, 96 patients obeying commands (motor response score of 6 to the GCS), 783 patients with OHCA due to external causes, and 2030 patients with no PaCO_2_ data in the ICU, a total of 607 patients were eligible for the final analysis. Of these, 53 (8.7%) patients experienced severe hypocapnia, 206 (33.9%) patients experienced mild hypocapnia, 96 (15.8%) patients experienced normocapnia, 98 (16.1%) patients experienced mild hypercapnia, 88 (14.5%) patients experienced severe hypercapnia, and 66 (10.9%) patients experienced both hypocapnia and hypercapnia (Figure 1).

### 3.2. Patients’ Characteristics and In-Hospital Data

Patients’ characteristics and pre- and in-hospital data of PaCO_2_ groups during the first 24 h after ROSC are shown in Table 1. A total of 225 (37.1%) patients had OHCA with shockable rhythm (VF/pVT) as the initial cardiac rhythm. The proportion of witnessed arrest was 61.6%. The proportion of OHCA patients who received bystander CPR was 35.7%, and CPR was performed for more than 10 min before ROSC in 70.8% of patients. Treatment using a mechanical circulatory device was performed in 113/607 (18.6%) patients. TTM was performed in 74.2% (167/225) of patients with shockable rhythm as the initial cardiac rhythm, in 30.5% (39/128) of patients with PEA, and in 33.7% (29/86) of patients with asystole (319 patients in total; 52.6%). There were significant differences in sex, initial cardiac rhythm, CPR duration, use of a mechanical circulatory device, hyperoxia exposure, and cause of cardiac arrest between the six groups (all *p*-value < 0.05 in Table 1).

### 3.3. PaCO_2_ Immediately after ROSC and PaCO_2_ upon Admission to the ICU

The relationship between PaCO_2_ immediately after ROSC and PaCO_2_ upon admission to the ICU is shown in Figure 2.

Severe hypercapnia was the most common state when measuring PaCO_2_ immediately after ROSC, followed by normocapnia, with severe hypocapnia being the least common (severe hypercapnia, 256; mild hypercapnia, 76; normocapnia, 111; mild hypocapnia, 79; severe hypocapnia, 15). PaCO_2_ upon admission to the ICU tended to show normocapnia (227 patients, 37.4%) regardless of PaCO_2_ immediately after ROSC. In the group that showed severe hypercapnia immediately after ROSC, severe hypercapnia was the second most common PaCO_2_ condition upon admission to the ICU. Similarly, patients with mild hypercapnia and mild hypocapnia in PaCO_2_ immediately after ROSC showed mild hypercapnia and mild hypocapnia as the second most common condition upon admission to the ICU. None of the patients who presented with severe hypocapnia immediately after ROSC had hypercapnia upon admission to the ICU.

We found that 27.6% (62/225) of patients with a shockable initial rhythm such as VF or pVT had severe hypercapnia immediately after ROSC, and 63.1% of patients with an unshockable initial rhythm such as PEA or asystole had severe hypercapnia. The CPR duration of patients with severe hypercapnia immediately after ROSC was longer than that of other groups, and the ratio of CPR > 10 min tended to be higher with statistical significance. Both poor neurological prognosis and mortality rates were highest in the severe hypercapnia group (Table 2).

### 3.4. Outcomes

The proportions of patients with 1-month poor neurological outcome were 83.0% (44/53), 68.0% (140/206), 51.0% (49/96), 46.9% (46/46.9), 75.0% (66/88), and 77.3% (51/66) of patients with severe hypocapnia exposure, mild hypocapnia exposure, normocapnia exposure, mild hypercapnia exposure, severe hypercapnia exposure, and exposure to both hypocapnia and hypercapnia, respectively.

After adjustment for potential confounders, severe hypocapnia, mild hypocapnia, severe hypercapnia, and exposure to both hypocapnia and hypercapnia were more likely to have a 1-month poor neurologic status than those with mild hypercapnia (with mild hypercapnia exposure as a reference, adjusted ORs [95% CI] for 1-month poor neurologic status were 6.68 [2.16–20.67], 2.56 [1.30–5.04], 2.62 [1.06–6.47], 5.63 [2.21–14.34], respectively; Table 3A). There was no significant difference in 1-month poor neurologic status between those with normocapnia and those with mild hypocapnia.

Patients with exposure to both hypocapnia and hypercapnia had the highest 1-month mortality rate after adjustment for potential confounders. However, there was no significant difference among the hypocapnia, normocapnia, or hypercapnia exposure groups (Table 3B).

In addition, the result of sensitivity analysis for OHCA 494 patients without treatment using a mechanical circulatory device was comparable to the main analysis (Table 1, Appendix A).

## 4. Discussion

In this multicenter retrospective observational study of 607 OHCA patients in the JAAM-OHCA registry, we evaluated whether PaCO_2_ 24 h after ROSC was associated with 1-month poor neurological outcome or 1-month mortality after ROSC.

In the multivariable logistic regression analysis, we demonstrated that mild hypercapnia exposure (45–55 mmHg) was associated with a lower risk of 1-month poor neurological outcome when compared with the outcome for hypocapnia (<25 mmHg or 25–35 mmHg), severe hypercapnia (>55 mmHg), or exposure to both hypocapnia and hypercapnia; however, there was no significant difference between the outcomes for normocapnia (35–45 mmHg) and mild hypercapnia exposure. Moreover, we also found that patients with exposure to both hypocapnia and hypercapnia within 24 h after ROSC could potentially have high 1-month mortality.

Our finding is consistent with the recommendation of international resuscitation guidelines and a meta-analysis by McKenzie et al. [7], which suggested that normocapnia exposure was also associated with better clinical outcomes. In addition, our findings suggest that exposure to mild hypercapnia may contribute to improved neurological prognosis.

The results of observational studies of hypercapnia after ROSC and neurological outcomes have been contradictory. Wang et al. [21] reported that the likelihood of a favorable neurological outcome decreases with increasing levels of PaCO_2_. Roberts et al. [4] reported that patients with hypercapnia (>50 mmHg) had a worse neurological outcome than patients with normocapnia (30–50 mmHg). Schneider et al. [9] reported that the hypercapnia (>45 mmHg) group had a higher rate of discharge home among survivors compared with the normocapnia (35–45 mmHg) group. Vaahersalo et al. [8] reported that hypercapnia (>45 mmHg) was associated with a good neurological outcome at 12 months after ROSC from OHCA. An experimental study in an immature rat model by Vannucci et al. [22] showed that normocapnic cerebral hypoxia-ischemia is associated with less severe brain damage than hypocapnic hypoxia-ischemia and that mild hypercapnia is more protective than normocapnia. Based on these results, the randomized Carbon Control and Cardiac Arrest pilot trial was performed by Eastwood et al. [10] and showed that targeted therapeutic mild hypercapnia (TTMH) after cardiac arrest during the first 24 h was safely feasible and that TTMH patients had a pattern of improved global functional outcome at six months. Our findings suggest that normocapnia or mild hypercapnia exposure after ROSC is associated with better neurological outcomes and may provide scientific evidence for further study including randomized controlled trials to determine the optimal target therapeutic PaCO_2_ level during the initial period after ROSC. The Targeted Therapeutic Mild Hypercapnia After Resuscitated Cardiac Arrest (TAME) trial will determine whether targeted therapeutic mild hypercapnia applied during the first 24 h of mechanical ventilation in the ICU improves neurological outcome at 6 months compared to standard care targeted normocapnia. The trial is a phase III multicenter randomized controlled trial in resuscitated OHCA patients and is estimated to be completed in December 2022 [23].

It is still unknown when PaCO_2_ levels should be started to be controlled by ventilation to improve the neurological outcomes of OHCA patients. The ventilation strategy to improve neurological prognosis during CPR and the ventilation strategy to avoid positive-pressure hyperventilation for achieving ROSC are not necessarily the same, making it difficult to strictly manage patients’ ventilation by targeting a certain PaCO_2_ level during CPR. PaCO_2_ immediately after ROSC is likely to affect the initial PaCO_2_ after the ICU admission. The rate of normocapnia at the ICU admission was only 37.4% even with post-ROSC ventilation therapy; that is, even though some guidelines currently recommend management for normocapnia, the range of PaCO_2_ management in the first 24 h after ROSC in our study was wide. Thus, it is important to stabilize the cardiovascular system and to intervene immediately after ROSC in the emergency room to target mild hypercapnia and normocapnia with appropriate ventilation therapy.

We found that patients with severe hypercapnia immediately after ROSC had poor 1-month neurological outcomes and high 1-month mortality than any other PaCO_2_ level. However, since PaCO_2_ values immediately after ROSC including severe hypercapnia are the result of various factors such as the cause of cardiopulmonary arrest, physiologic effects of cardiac arrest or CPR, and respiratory complications, they may be referred to as a prognostic indicator rather than an index for post-cardiac arrest care.

Despite the importance of our findings, this study has several limitations. First, this study had a retrospective design. It has the inherent limitations of possible bias due to residual confounding factors and missing data. Some information on during of mechanical ventilation, mode of mechanical ventilation, drugs administered including catecholamines and sedatives could not be obtained. The timing of the ABG test used in the analysis was strictly defined to reduce the risk of measurement bias. However, three blood gas analysis data points were used in the analysis, which may not be sufficient to assess the time course of PaCO_2_ in the 24-h intensive care management after ROSC. Second, the exact incidence rates of poor neurological outcome and mortality are still uncertain because consecutive OHCA patients who survived more than 24 h after ROSC were included to analyze enough ABG data. However, we believe that it is clinically valuable to examine the effect of the difference in PaCO_2_ levels on neurological patient outcomes in patients who underwent post-cardiac arrest care for more than 24 h after ROSC. Third, we adjusted for the cause of cardiac arrest defined by each physician’s diagnosis or the Utstein style classification [24]; however, the accuracy of the causal classification of cardiac arrest is unknown. In particular, the classification of “cardiac cause excluding acute coronary syndrome (presumed cardiac cause)” in cases of unexplained cardiac arrest may be more common than it should be. We used a large sample size to minimize the potential sources of biases. Finally, our study was conducted solely using data from Japan. Further studies including multiple medical centers in different countries will provide more generalized information and external validity. Therefore, these limitations should be considered, and further studies are warranted to assess the validity of our findings.

## 5. Conclusions

We conclude that compared to hypocapnia or severe hypercapnia exposure, normocapnia and mild hypercapnia exposure during the first 24 h after ROSC is associated with better neurological outcomes in OHCA patients. Further studies are warranted to determine which PaCO_2_ levels should be targeted to improve outcomes and how ventilation of OHCA patients should be managed.

## Figures and Tables

**Figure 1 jcm-11-01523-f001:**
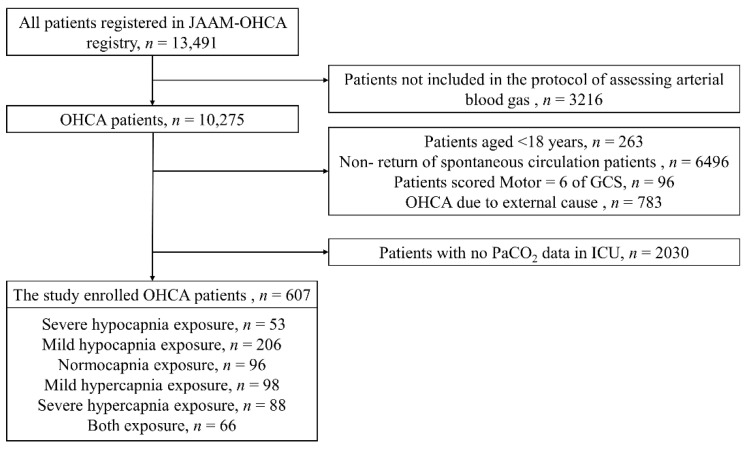
Flowchart of included and excluded patients of the study. GCS, Glasgow coma score; ICU, intensive care unit; JAAM, Japanese Association for Acute Medicine; OHCA, out-of-hospital cardiac arrest; PaCO_2_, partial pressure of arterial carbon dioxide.

**Figure 2 jcm-11-01523-f002:**
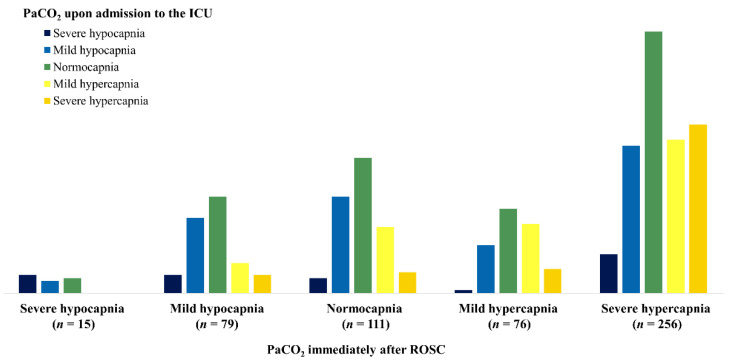
The relationship between PaCO_2_ immediately after ROSC and PaCO_2_ upon admission to the ICU. Each PaCO_2_ level immediately after ROSC indicates what level of PaCO_2_ will be reached upon subsequent admission to the ICU. ICU, intensive care unit; PaCO_2_, partial pressure of arterial carbon dioxide; ROSC, return of spontaneous circulation.

**Table 1 jcm-11-01523-t001:** Patients’ characteristics of this study by PaCO_2_ groups during the first 24 h after the return of spontaneous circulation.

	PaCO_2_ Group												*p*
	(*n* = 607)												
	Severe Hypocapnia Exposure		Mild Hypocapnia Exposure		Normocapnia Exposure		Mild Hypercapnia Exposure		Severe Hypercapnia Exposure		Both Exposure		
	(*n* = 53)		(*n* = 206)		(*n* = 96)		(*n* = 98)		(*n* = 88)		(*n* = 66)		
Male	35	(66.0)	144	(69.9)	75	(78.1)	84	(85.7)	63	(71.6)	52	(78.8)	0.029
Age, years	66	(56–73)	65	(54–75)	67	(54–75)	66	(54–77)	68	(56–79)	62	(48–71)	0.216
Initial cardiac rhythm													0.005
VF/pVT	20	(37.7)	79	(38.3)	42	(43.8)	42	(42.9)	20	(22.7)	22	(33.3)	
PEA	11	(20.8)	45	(21.8)	20	(20.8)	10	(10.2)	24	(27.3)	18	(27.3)	
Asystole	10	(18.9)	19	(9.2)	7	(7.3)	16	(16.3)	25	(28.4)	9	(13.6)	
Other	4	(7.5)	19	(9.2)	7	(7.3)	5	(5.1)	5	(5.7)	5	(7.6)	
Witnessed arrest	32	(60.4)	127	(61.7)	61	(63.5)	55	(56.1)	56	(63.6)	43	(65.2)	0.871
Bystander performed CPR	17	(32.1)	69	(33.5)	38	(39.6)	41	(41.8)	33	(37.5)	19	(28.8)	0.309
CPR duration, min	31	(22–47)	20	(11–35)	11	(5–26)	16	(6–34)	18	(11–30)	21	(9–30)	<0.001
CPR > 10 min	39	(73.6)	115	(55.8)	38	(39.6)	44	(44.9)	54	(61.4)	37	(56.1)	<0.001
GCS at hospital arrival	3	(3–3)	3	(3–3)	3	(3–3)	3	(3–3)	3	(3–3)	3	(3–3)	0.061
Mechanical circulatory device	21	(39.6)	52	(25.2)	17	(17.7)	13	(13.3)	2	(2.3)	8	(12.1)	<0.001
TTM	21	(39.6)	108	(52.4)	58	(60.4)	56	(57.1)	41	(46.6)	35	(53.0)	0.155
Hyperoxia exposure	32	(60.4)	83	(40.3)	47	(49.0)	32	(32.7)	34	(38.6)	20	(30.3)	0.005
Cause of cardiac arrest													<0.001
ACS	20	(37.7)	66	(32.0)	37	(38.5)	36	(36.7)	10	(11.4)	20	(30.3)	
Cardiac cause excluding ACS (presumed cardiac cause)	23	(43.4)	98	(47.6)	47	(49.0)	46	(46.9)	38	(43.2)	25	(37.9)	
Respiratory cause	0	(0)	9	(4.4)	4	(4.2)	7	(7.1)	26	(29.5)	7	(10.6)	
Cerebrovascular cause	3	(5.7)	11	(5.3)	3	(3.1)	3	(3.1)	6	(6.8)	8	(12.1)	
Malignant tumor	1	(1.9)	2	(1.0)	0	0.0	2	(2.0)	1	(1.1)	0	(0)	
Others or unknown	6	(11.3)	20	(9.7)	5	(5.2)	4	(4.1)	7	(8.0)	6	(9.1)	

Values are presented as *n* (%) or median (interquartile range: quartile 1–quartile 3). ACS, acute coronary syndrome; CPC, cerebral performance category; CPR, cardiopulmonary resuscitation; GCS, Glasgow coma scale; IQR, interquartile range; PEA, pulseless electric activity; pVT, pulseless ventricular tachycardia; TTM, targeted temperature management; VF, ventricular fibrillation.

**Table 2 jcm-11-01523-t002:** Patients’ characteristics and outcomes of this study by PaCO_2_ immediately after return of spontaneous circulation.

			PaCO_2_ Level										*p*
			(*n* = 607)										
	Missing		Severe Hypocapnia		Mild Hypocapnia		Normocapnia		Mild Hypercapnia		Severe Hypercapnia		
	(*n* = 70)		(*n* = 15)		(*n* = 79)		(*n* = 111)		(*n* = 76)		(*n* = 256)		
Male	51	(72.9)	14	(93.3)	64	(81.0)	84	(75.7)	54	(71.1)	186	(72.7)	0.337
Age, years	65	(50–72)	69	(56–81)	67	(56–74)	63	(54–72)	63	(49–72)	67	(56–79)	0.029
Initial cardiac rhythm													<0.001
VF/pVT	29	(41.4)	8	(53.3)	42	(53.2)	47	(42.3)	37	(48.7)	62	(24.2)	
PEA	13	(18.6)	3	(20.0)	11	(13.9)	12	(10.8)	10	(13.2)	79	(30.9)	
Asystole	7	(10.0)	0	0.0	6	(7.6)	12	(10.8)	5	(6.6)	56	(21.9)	
Other	6	(8.6)	2	(13.3)	6	(7.6)	11	(9.9)	6	(7.9)	14	(5.5)	
Witnessed arrest	42	(60.0)	11	(73.3)	49	(62.0)	65	(58.6)	49	(64.5)	158	(61.7)	0.629
Bystander performed CPR	22	(31.4)	7	(46.7)	34	(43.0)	40	(36.0)	25	(32.9)	89	(34.8)	0.577
CPR duration, min	28	(13–50)	13	(7–51)	11	(5–26)	14	(6–34)	13	(6–26)	23	(13–32)	<0.001
CPR > 10min	43	(61.4)	7	(46.7)	31	(39.2)	46	(41.4)	33	(43.4)	167	(65.2)	<0.001
GCS score at hospital arrival	3	(3–3)	3	(3–3)	3	(3–3)	3	(3–4)	3	(3–3)	3	(3–3)	<0.001
1-month poor neurologic status ^a^	45	(64.3)	8	(53.3)	34	(43.0)	53	(47.7)	41	(53.9)	215	(84.0)	<0.001
1-month mortality	28	(40.0)	5	(33.3)	22	(27.8)	21	(18.9)	26	(34.2)	125	(48.8)	<0.001

Values are presented as *n* (%) or median (interquartile range: quartile 1–quartile 3). CPR, cardiopulmonary resuscitation; GCS, Glasgow coma scale; PEA, pulseless electric activity; VF, ventricular fibrillation; pVT, pulseless ventricular tachycardia. ^a^ Poor neurologic status defined as Cerebral Performance Category ≥ 3.

**Table 3 jcm-11-01523-t003:** Outcomes.

(A) Association with 1-Month Poor Neurologic Status and Exposure to PaCO_2_ of 24 h Post-Return of Spontaneous Circulation.							
	Total	1-month poor neurologic status *^a^* (%)		Crude OR	(95% CI)	Adjusted *^b^* OR	(95% CI)
Severe hypocapnia exposure	53	44	(83.0)	5.53	(2.44–12.5)	6.68	(2.16–20.67)
Mild hypocapnia exposure	206	140	(68.0)	2.40	(1.46–3.93)	2.56	(1.30–5.04)
Normocapnia exposure	96	49	(51.0)	1.20	(0.68–2.12)	1.77	(0.81–3.86)
Mild hypercapnia exposure	98	46	(46.9)	Reference		Reference	
Severe hypercapnia exposure	88	66	(75.0)	3.39	(1.82–6.33)	2.62	(1.06–6.47)
Both exposure	66	51	(77.3)	3.84	(1.91–7.73)	5.63	(2.21–14.34)
**(B) Association with 1-Month Mortality and Exposure to PaCO_2_ of 24 h Post-Return of Spontaneous Circulation.**							
	Total	1-month mortality (%)		Crude OR	(95% CI)	Adjusted *^b^* OR	(95% CI)
Severe hypocapnia exposure	53	25	(47.2)	2.01	(1.08–3.72)	1.29	(0.52–3.21)
Mild hypocapnia exposure	206	82	(39.8)	1.83	(1.08–3.11)	1.65	(0.84–3.25)
Normocapnia exposure	96	23	(24.0)	0.88	(0.46–1.70)	1.15	(0.51–2.62)
Mild hypercapnia exposure	98	26	(26.5)	Reference		Reference	
Severe hypercapnia exposure	88	37	(42.0)	2.47	(1.23–4.98)	1.30	(0.57–2.94)
Both exposure	66	34	(51.5)	2.94	(1.52–5.69)	3.06	(1.34–6.99)

CI, confidence interval; OR, odds ratio. *^a^* Poor neurologic status defined as Cerebral Performance Category ≥ 3. *^b^* Adjusted for sex, age, witnessed arrest, bystander performed cardiopulmonary resuscitation, initial cardiac rhythm, cardiopulmonary resuscitation duration >10 min, Glasgow coma score at hospital arrival, hyperoxia exposure, mechanical circulatory device, targeted temperature management, cause of cardiac arrest, PaCO_2_ immediately after return of spontaneous circulation, PaCO_2_ group.

## Data Availability

The data and analysis scripts in this study are available upon request from the corresponding author.

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
