# Peer review of "Post-Resuscitation Partial Pressure of Arterial Carbon Dioxide and Outcome in Patients with Out-of-Hospital Cardiac Arrest: A Multicenter Retrospective Cohort Study"

_jcm, 2022, doi:10.3390/jcm11061523_

Round 1
Reviewer 1 Report
Dear Authors,
You have provided a well written and presented research paper on the partial pressure of carbon dioxide in resuscitated cardiac arrest patients from 73 institutions.
The background, aim and methods are adequately described, understanding the methodology and nature of retrospective database projects.
The results section is initially difficult to interpret given the 5 CO2 groups. Further columns of the CO2-groups as shown Figure 2. are hard to appreciate when printed in black & white.
Recommendation
For the results section - order the findings as 1st patient characteristics, 2nd PaCO2 exposure and immediate post-ROSC, 3rd outcomes. As this is chronological.
I found Table 2 difficult to interpret given the top half and bottom half approach. Please include a line to reference which association is being presented.
Do you have and could you please present information on the time-course of PaCO2 in the intensive care. Meaning that it would be beneficial to see how many arterial blood gases were taken during ICU care in the first 24 h and when the peak PaCO2 was experienced.
Do you have any information on during of mechanical ventilation and ICU supports? Such information would be beneficial to know as PaCO2 targeting or management is easier when intubated.
Best wishes
Reviewer
Author Response
Dear Reviewer 1,
Thank you for your thorough reviews of Manuscript ID jcm-1598368 entitled "Post-resuscitation partial pressure of arterial carbon dioxide and outcome in patients with out-of-hospital cardiac arrest: a multicenter retrospective cohort study".
We have carefully read your comments and revised our manuscript accordingly. Thank you in advance for your further consideration of this manuscript. We hope that this revised version will now be considered acceptable for publication.
Best wishes,
Authors
Point 1: The results section is initially difficult to interpret given the 5 CO2 groups. Further columns of the CO2-groups as shown Figure 2. are hard to appreciate when printed in black & white.
Response 1: Thank you for your thorough review and important advice. One of our goals was to evaluate the CO2 values in a more fine-grained way than in the previous literature. As you pointed out, I think it is difficult to interpret given the 5 CO2 groups. I tried to print the graph in color to make it as easy to understand as possible.
Point 2: For the results section - order the findings as 1st patient characteristics, 2nd PaCO2 exposure and immediate post-ROSC, 3rd outcomes. As this is chronological.
Response 2: Thank you for your appropriate suggestions. I have replaced the order of the results sections as follows; 1st population, 2nd patient characteristics, 3rd PaCO2 exposure and immediate post-ROSC, 4th outcomes(As this is chronological.). This has also changed the numbering of tables.
Point 3: I found Table 2 difficult to interpret given the top half and bottom half approach. Please include a line to reference which association is being presented.
Response 3: Thank you for your important advice and suggestions. In the revision at point 2, table2 has been changed to table3. I've added a line to show the top half and bottom half approach, and a sub-title to distinguish between top and bottom. In line with this, I have also distinguished table3(A) and table3(B) in the text.
Point 4: Do you have and could you please present information on the time-course of PaCO2 in the intensive care. Meaning that it would be beneficial to see how many arterial blood gases were taken during ICU care in the first 24 h and when the peak PaCO2 was experienced.
Response 4: Thank you for your valuable question and opinion. I think too that almost all patients in ICU actually have had more than 2 or 3 blood gas tests. Ideally, all of this blood gas test information should be used to infer and group the actual time-course of PaCO2. However, the protocol in our registry only included four points of ABG information: before ROSC, immediately after ROSC, at ICU admission and 24 hours after ROSC. This is an important limitation, and we have added the following to the discussion section (limitations); Three blood gas analysis data points were used in our analysis, which may not be sufficient to assess the time course of PaCO2 in the 24-hour intensive care management after ROSC.
Point 5: Do you have any information on during of mechanical ventilation and ICU supports? Such information would be beneficial to know as PaCO2 targeting or management is easier when intubated.
Response 5: In Our study protocol, some information on during of mechanical ventilation, mode of mechanical ventilation, drugs administered including catecholamines and sedatives could not be obtained. As with Response 5, we have included it as a limitation.
Reviewer 2 Report
Thank you for opportunity to review this manuscript. It is retrospective observation multicenter study and main conclusion is association between normo- and mild hypercapnia 24 hours after ROSC with better neurological outcome. My notices are presented bellow.
General comments:
- 607 patients represents large cohort of OHCA population
- Primary and secondary outcomes are clear formulated
- English language level is sufficient and intelligible
- PaCO2 level in postresuscitation care is resonable question with still unclear management
- Study was approved by ethics comitee
Introduction:
I cant identify clearly which new information compared to previous studies your study design can bring from presented overview.
Last paragraph meaning is unclear, summary of results ( possible relation with study hypothesis formulation) are not presented.
Methods:
It is known, if PaCO2 levels in any participating centers were targeting to any level according to local protocol or any interventional study?
How were PaCO2 levels immediately after ROSC provided? ( It is different methodology in prehospital care?)
Results:
More than 2000 pts excluded for poor data, it is known reason? ( I expect unified CRF for registr) and patients selection could lead to signifficant biases.
Bystander CPR level is low compared to known data, can you comment it?
Table 1 is for me unclear- differences between unique groups and more p levels in one column
There is very high level of ECLS (18,6%)- it is extraordinary and it seems to be selected population ( ECLS patients have sufficient PaCO2 levels information and therefore were included) and this therapy is more efective compared to mechanical ventilation, however their prognosis is very poor. I am not sure if such patients population can be included to analysis.
Under figure 2, last paragraph on page 6: …hypercapnia severe hypercapnia….. ?
Table 3-p levels are related to which variables?
Discussion:
It is good summary and comparing with revious data, however would be great to identify new additional information resulted from this study which were no presented in previous studies
Author Response
Dear Reviewer 2,
Thank you for your thorough reviews of Manuscript ID jcm-1598368 entitled "Post-resuscitation partial pressure of arterial carbon dioxide and outcome in patients with out-of-hospital cardiac arrest: a multicenter retrospective cohort study".
We have carefully read your comments and revised our manuscript accordingly. Thank you in advance for your further consideration of this manuscript. We hope that this revised version will now be considered acceptable for publication.
Best wishes,
Authors
Point 1: Introduction
I cant identify clearly which new information compared to previous studies your study design can bring from presented overview. Last paragraph meaning is unclear, summary of results ( possible relation with study hypothesis formulation) are not presented.
Response 1: Thank you for your thorough review and important opinion.
Accordingly, we revised the last paragraph into the following ​description: “Although there are studies that have classified PaCO2 values at 24-h after ROSC into three categories (hypocapnia, normocapnia, and hypercapnia) and evaluated their association with neurological outcomes, there are few randomized controlled trials and observational studies that have finely classified PaCO2 values and evaluated in a more detailed. This study aimed to estimate the target range of PaCO2 exposure with the better outcome by dividing patients with ROSC from OHCA into six categories according to the PaCO2 values they were exposed to. ”
Point 2 : Methods
It is known, if PaCO2 levels in any participating centers were targeting to any level according to local protocol or any interventional study?
Response 2: Thank you for your question. It is not clear whether the targeting PaCO2 levels in each participating center were set by local protocols or intervention study. At least, the protocol of our registry did not specify a target PaCO2 level.
Point 3: Methods
How were PaCO2 levels immediately after ROSC provided? ( It is different methodology in prehospital care?)
Response 3: Thank you for your question. PaCO2 levels immediately after ROSC were in-hospital data. In patients with pre-hospital ROSC, the data of PaCO2 levels immediately after ROSC used the first PaCO2 values obtained after hospital arrival or treated them as missing.
The 'Data Collection' section has been revised to avoid ambiguity as follows; Data of PaCO2 values were collected three times; immediately after ROSC, upon admission to the intensive care unit [ICU], and 24-h after ROSC. In patients with pre-hospital ROSC, the data of PaCO2 immediately after ROSC used the first PaCO2 values obtained after hospital arrival or treated them as missing.
Point 4: Results
More than 2000 pts excluded for poor data, it is known reason? ( I expect unified CRF for registr) and patients selection could lead to signifficant biases.
Response 4: Patients were excluded if they were missing at least one of the ABG data at ICU admission or 24 hours after ROSC. The exact reason for missing is unclear, however, I guess that the cases for which blood gas data (PaCO2 data) were not available include many patients who died within 24 hours, did not qualify for active intensive care, or who withdrew from treatment. The reasons are the differences in outcomes, as follows;
2030 patients excluded due to missing data; 1-month survival rate was 14.5%(294/2030) and patients with a good neurological outcome (CPC=1,2) were 7.8%(159/2030)
607 patients included in the analysis; 1-month survival rate was 62.6%(380/607) and patients with a good neurological outcome (CPC=1,2) were 34.8%(211/607)  
Point 5: Results
Bystander CPR level is low compared to known data, can you comment it?
Response 5: Thank you for your valuable comments. 35.7ï¼…(217/607). In our study, the bystander CPR rate means the rate of citizen CPR. The OHCA patients included in the analysis include those who developed CPA after contact with the EMS, which may have resulted in a larger denominator and therefore a lower rate. (citizen CPR/citizen bystander vs. citizen CPR/[citizen bystander + EMS witnessing CPA))
Point 6: Results
Table 1 is for me unclear- differences between unique groups and more p levels in one column
Response 6: We are analyzing to see if there is any difference anywhere between any of the 6 groups. I would be happy to delete the p-value in the Main text, Tables 1 and 3(2 after changes) If the editor preferred to do so.
Point 7: Results
There is very high level of ECLS (18,6%)- it is extraordinary and it seems to be selected population ( ECLS patients have sufficient PaCO2 levels information and therefore were included) and this therapy is more efective compared to mechanical ventilation, however their prognosis is very poor. I am not sure if such patients population can be included to analysis.
Response 7: Thank you for your valuable comments. Recent studies have shown that not all patients have a poor prognosis, depending on the indication for ECLS [Refe]. In the population we analysed, with ECLS vs. without ECLS; had 45.1% vs. 35.7 for 1-month mortality(p=0.06), had 70.8% vs. 64.1% for 1-month poor neurologic status(p=0.177). There was no significant difference in prognosis, but the frequency of poor prognosis was higher in patients with ECLS. Hence, we conducted a sensitivity analysis for all OHCA patients included in the study, after excluding patients with treatment using a mechanical circulatory device. The method has been described and the results added. Table S1 is included in the Supplementary Material. The results were comparable to the main analysis.
References:Yannopoulos, D., Bartos, J., Raveendran, G., Walser, E., Connett, J., Murray, T. A., Collins, G., Zhang, L., Kalra, R., Kosmopoulos, M., John, R., Shaffer, A., Frascone, R. J., Wesley, K., Conterato, M., Biros, M., Tolar, J., & Aufderheide, T. P. (2020). Advanced reperfusion strategies for patients with out-of-hospital cardiac arrest and refractory ventricular fibrillation (ARREST): a phase 2, single centre, open-label, randomised controlled trial. Lancet (London, England), 396(10265), 1807–1816. https://doi.org/10.1016/S0140-6736(20)32338-2
Point 8: Results
Under figure 2, last paragraph on page 6: …hypercapnia severe hypercapnia….. ?
Response 8: Thank you for pointing this out. It was our mistake. We have corrected it.
Point 9: Results
Table 3-p levels are related to which variables?
Response 9: We are analyzing to see if there is any difference anywhere between any of the 6 groups. I would be happy to delete the p-value in the Main text, Tables 1 and 3 (2 after changes) If the editor preferred to do so.
Point 10: Discussion
It is good summary and comparing with revious data, however would be great to identify new additional information resulted from this study which were no presented in previous studies
Response 10: Thank you for your important advice. The new additional information resulted from this study is limited, but what is new in our study is that we have classified PaCO2 values in more detail and treated PaCO2 immediately after ROSC and PaCO2 during ICU management separately. I hope our study helps future research.
Round 2
Reviewer 2 Report
Thank you to authors for their answers and corrections.
I have only one comment to table 1 and 3 and p-values which are not clear for me, but i expect editor will make a decision.
Thank you again for your work and kind answers to my comments.
Author Response
Dear Reviewer 2,
Thank you for your comment. Regarding table 1 and 3 and p-values, If the editors have any comments, I will try to follow their decision. I thank you once again for your careful peer review, which has been a great help to me.
Author